# ShortcutBreaker: Low-Rank Noisy Bottleneck with Global Perturbation Attention for Multi-Class Unsupervised Anomaly Detection

## Abstract

Multi-class unsupervised anomaly detection (MUAD) has garnered growing research interest, as it seeks to develop a unified model for anomaly detection across multiple classes—eliminating the need to train separate models for distinct objects and thereby saving substantial computational resources. Under the MUAD setting, while advanced Transformer-based architectures have brought significant performance improvements, identity shortcuts persist: they directly copy inputs to outputs, narrowing the gap in reconstruction errors between normal and abnormal cases, and thereby making the two harder to distinguish. Therefore, we propose ShortcutBreaker, a novel unified feature-reconstruction framework for MUAD tasks, featuring two key innovations to address the issue of shortcuts. First, drawing on matrix rank inequality, we design a low-rank noisy bottleneck (LRNB) to project high-dimensional features into a low-rank latent space, and theoretically demonstrate its capacity to prevent trivial identity reproduction. Second, leveraging ViT's global modeling capability instead of merely focusing on local features, we incorporate a global perturbation attention to prevent information shortcuts in the decoders. Extensive experiments are performed on four widely used anomaly detection benchmarks, including three industrial datasets (MVTec-AD, ViSA, and Real-IAD) and one medical dataset (Universal Medical). The proposed method achieves a remarkable image-level AUROC of *99.8%, 98.9%, 90.6%*, and *87.8%* on these four datasets, respectively, consistently outperforming previous MUAD methods across different scenarios. Our code will be released.

## 1 Introduction

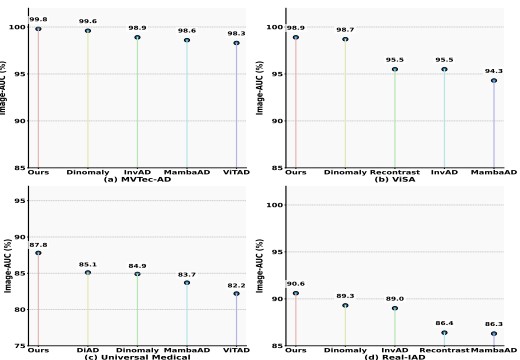

Figure 1: Performance comparison in terms of image-level AUC on MVTec, ViSA , Universal Medical and Real-IAD .

Anomaly detection in industrial and medical imaging aims to identify abnormal patterns among normal cases, saving labor and time in collecting and labeling anomalies. Given the abundance of normal cases and the scarcity of anomalies, this task is typically tackled via an unsupervised paradigm using only normal training samples. Before deep learning, anomaly detection relied on traditional techniques: density-based methods (Breunig et al., 2000; Guan et al., 2015), distance-based methods (Knorr et al., 2000; Angiulli et al., 2005), and statistics-based methods (Hido et al., 2011; Rousseeuw & Hubert, 2011). Current state-of-the-art methods employ deep learning networks pre-trained on ImageNet (Deng et al., 2009) to capture discriminative features. Feature reconstruction methods (Guo et al., 2023; 2024; Deng & Li, 2022) reconstruct encoder-extracted features, assuming accurate reconstruction of normal regions but failure on unseen anomalies. Memory matching methods (Yi & Yoon, 2020; Defard

et al., 2021; Roth et al., 2022) memorize training-set normal features for inference matching, with those in (Defard et al., 2021; Roth et al., 2022) using pre-trained encoders for discriminative features. Pseudo-anomaly methods (Liu et al., 2023; Li et al., 2021; Zavrtanik et al., 2021) convert UAD to a supervised task by generating pseudo anomalies via noise addition to normal images/features. Hybrid methods (Tien et al., 2023; Zhao et al., 2023) integrate normalizing flows (Zhao et al., 2023) or pseudo noise (Tien et al., 2023) into feature reconstruction for UAD.

Despite their success, these methods are limited to a one-model-one-class setup, requiring substantial storage for per-class models—especially with many disease types (You et al., 2022). To address this, UniAD and follow-ups propose unified models for multi-class unsupervised anomaly detection (MUAD). However, identity mapping often emerges here, where the model returns input copies regardless of normality, enabling effective reconstruction of even anomalous samples and hindering detection (You et al., 2022).

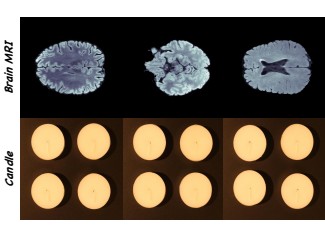
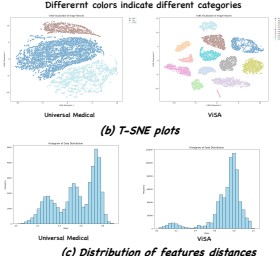

Over the past three years, substantial progress has been made in multi-class unsupervised anomaly detection (MUAD), with explorations into pretrained vision transformers (ViTs) (Zhang et al., 2023; Guo et al., 2025), state space models (Mambas) (He et al., 2024a), diffusion models (Yin et al., 2023; He et al., 2024b), and other approaches (Guo et al., 2024; Zhao, 2023; He et al., 2024c). Nevertheless, the multi-class setting inevitably induces identity mapping in most methods, resulting in performance degradation. While previous studies (You et al., 2022; Guo et al., 2025) have endeavored to mitigate shortcut learning, the efficacy of their proposed techniques in complex and highly diverse scenarios remains limited (see Fig. 1 (c)-(d)). This is particularly evident in

Figure 2: The visual patterns in the medical field are richer than those in the industrial field. (a) Visualization results, (b) T-SNE plots (Van der Maaten & Hinton, 2008), and (c) distributions of feature distances for the Universal Medical and ViSA datasets (Zou et al., 2022) are presented. These experimental results help observe the diversity of each dataset. For T-SNE plots, we extract the final features from the pre-trained ResNet-50 (He et al., 2016). Moreover, we use LPIPs (Zhang et al., 2018) to compare feature distances between individual image pairs.

datasets such as Real-IAD and Universal Medical, where the abundance of normal patterns exacerbates the aforementioned issue (You et al., 2022). The effectiveness of simple noise operations (You et al., 2022; Guo et al., 2025) is diminished due to the enhanced noise robustness acquired through exposure to diverse visual patterns. Additionally, the neighbor mask attention mechanism (You et al., 2022) is specifically tailored for features locally extracted by CNNs, and is not suitable for the global modeling structure - Transformer. Consequently, to enhance performance, our work aims to efficiently address the problem of identity shortcuts. Furthermore, why are the Real-IAD and Universal Medical datasets more complex? We elaborate on this as follows: It is apparent that in comparison to widely adopted industrial datasets—MVTec-AD (encompassing 15 classes) (Bergmann et al., 2019) and ViSA (comprising 12 classes) (Zou et al., 2022)—Real-IAD (Wang et al., 2024) exhibits more diverse normal patterns, featuring 30 object categories and 5 camera views. In contrast to standardized industrial images, although the Universal Medical dataset (He et al., 2024b) consists of only 3 categories, the inherent heterogeneity within medical normal samples enriches the visual features (*e.g.*, variations in organ size and shape across different demographics, as illustrated in Fig. 2(a)). Moreover, the Universal Medical dataset displays a sparser t-SNE plot and a wider dispersion of feature distances compared to ViSA (see Fig. 2 (b) and (c)), indicating a higher degree of intrinsic diversity within medical data.

In this paper, we propose a simple yet effective framework for the MUAD task, named Shortcut-Breaker, built on the advanced DINO-pretrained vision transformer with two key innovations. First, drawing on two observed properties of low-rank matrix decomposition, we design a low-rank noisy bottleneck (LRNB) to effectively mitigate the identity mapping issue. Within LRNB, the low-rank property theoretically circumvents shortcut learning of unseen patterns, while learnable matrix parameters are optimized to reconstruct normal patterns. Second, considering that ViT-extracted fea-

tures share similar global information, we introduce a global perturbation attention (GPA) mechanism to curb information leakage from input to output in the decoder. In GPA, a global redistribution operation and a global-self-masking mechanism force the decoder to learn reconstruction based on longer-range and impaired features. To validate the effectiveness of the proposed method, we conduct extensive experiments on four publicly available datasets: MVTec-AD (Bergmann et al., 2019), ViSA (Zou et al., 2022), Universal Medical (He et al., 2024b), and Real-IAD (Wang et al., 2024). As presented in Fig. 1, our ShortcutBreaker achieves the highest image-level AUC: 99.8%, 98.9%, 88.2%, and 90.6% on these four datasets, respectively. Notably, on the complex Universal Medical and Real-IAD datasets, it outperforms previous methods by a significant margin. In summary, our contributions are:

- We observe and simulate the properties of matrix decomposition to design a low-rank noisy bottleneck, efficiently suppressing identity mapping.

- We propose a global perturbation attention mechanism, which effectively prevents shortcut learning in the decoder via global redistribution and global-self-masking operations.

- Our ShortcutBreaker consistently outperforms previous methods across all four datasets, demonstrating enhanced robustness in diverse scenarios.

## 2 RELATED WORKS

### 2.1 SINGLE CLASS UAD

Recently, reconstruction mechanisms have dominated unsupervised anomaly detection (UAD) methods, detecting outliers by evaluating reconstruction errors between input and output pixels or features. These approaches include pixel reconstruction methods (Schlegl et al., 2017a;b) and feature reconstruction methods (Deng & Li, 2022; Guo et al., 2023; 2024). Pixel reconstruction methods aim to recreate normal images from scratch using a generative model framework. They assume that reconstructing normal samples will produce low reconstruction errors, while abnormal samples—differing from training data—will result in higher errors. Feature reconstruction methods suggest that latent features extracted from a pre-trained encoder provide more discriminative and robust representations compared to pixel-level reconstructions. Building on the success of such methods, many subsequent approaches (*e.g.*, EDC (Guo et al., 2023) and AE-FLOW (Zhao et al., 2023)) leverage this foundation to enhance anomaly detection. For instance, EDC employs an unfrozen pre-trained encoder, which is fine-tuned to adapt to the target domain while using a stop-gradient operation to retain important domain-relevant features. AE-FLOW further extends this idea by combining feature map reconstruction errors with distribution distances (measured via normalizing flows) as the final anomaly score. However, simply applying single-class methods as part of this one-class-one-model scheme is memory-intensive (especially as the number of classes grows) and ill-suited for scenarios with high intra-class diversity in normal samples, such as multi-class UAD (You et al., 2022).

### 2.2 MULTI CLASS UAD

UniAD (You et al., 2022) pioneers the field of multi-class unsupervised anomaly detection (MUAD), presenting a unified model capable of detecting anomalies across various classes. Most subsequent works have explored advanced modules to build better reconstruction models for MUAD. For example, LafitE (Yin et al., 2023) and DiAD (He et al., 2024b) further advance the MUAD task by leveraging the generative power of diffusion models to better capture anomalies across multiple classes. ViTAD (Zhang et al., 2023) and MambaAD (He et al., 2024a) develope feature reconstruction-based MUAD methods using recently advanced modules: Vision Transformer (Zhang et al., 2023) and State Space Model (He et al., 2024a), respectively. Few methods explicitly aim to address the identity mapping issue. UniAD (You et al., 2022) counteracts this through techniques like feature jittering and neighbor-masked attention. Dinomaly (Guo et al., 2025) employs components such as noisy bottleneck dropout to disrupt input feature replication. However, these noise operations have limited effectiveness in preventing shortcut learning when training on complex medical/industrial datasets (Universal Medical and Real-IAD), and neighbor-masked attention—specifically designed for CNN-extracted features—is not suitable for advanced ViT architectures (Guo et al., 2025; Zhang et al., 2023).

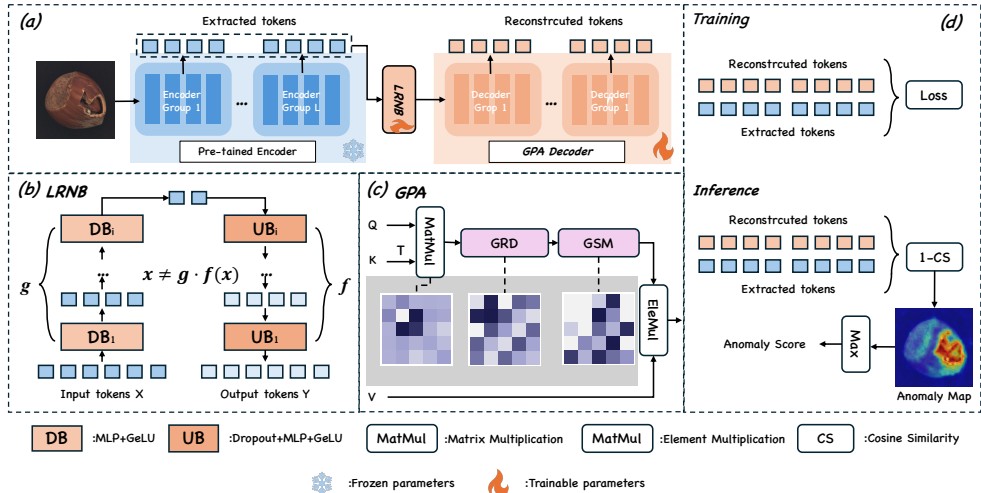

Figure 3: Flowchart of our ShortcutBreaker: (a) the overall structure of our proposed method; (b) the structure of the Low-Rank Noisy Bottleneck (LRNB); (c) the structure of the Global Perturbation Attention (GPA), which consists of GRD (global redistribution) and GSM (global-self masking); and (d) the training and inference pipelines.

## 3 METHOD

### 3.1 OVERVIEW

Following ViTAD and Dinomaly (Zhang et al., 2023; Guo et al., 2025), our ShortcutBreaker is a feature-reconstruction framework and constructed based on the vision transformer (ViT) structure. As depicted in Fig. 3(a), ShortcutBreaker consists of a pre-trained encoder, a bottleneck, and a decoder. The DINO-pretrained ViT model (Darcet et al., 2023) is utilized as the encoder, capturing informative feature tokens to facilitate subsequent reconstruction. The bottleneck is built upon a multi-layer perceptron (MLP), integrating multi-scale feature representations from intermediate encoder layers. The decoder maintains structural similarity with the encoder, employing transformer layers to reconstruct feature maps. As illustrated in Fig. 3(d), the encoder remains frozen during training, while the remaining components of ShortcutBreaker are optimized to reconstruct the encoder's feature tokens by minimizing inter-token representation discrepancies. During inference, this configuration enables accurate reconstruction of normal patterns but exhibits reconstruction failures in anomalous regions. The anomaly localization map is generated by computing 1-CS (where CS denotes cosine similarity) between the original and reconstructed feature tokens and then reshaping the result. The anomaly detection score is subsequently determined by the maximum value within this anomaly map.

### 3.2 LOW-RANK NOISY BOTTLENECK (LRNB)

As discussed in the introduction, under the MUAD setting, identity shortcuts readily occur, narrowing the gap in anomaly scores between normal and abnormal cases. Previous methods (Guo et al., 2025; You et al., 2022) propose perturbing extracted feature tokens to compel the network to reconstruct from information-impaired normal features, rather than directly copying them. However, these noisy operations are often limited in effectively preventing shortcut learning in complex scenarios. We hypothesize that richer visual patterns inherent in the data enhance the reconstruction model's robustness to noise. Consequently, we aim to design a novel paradigm for avoiding shortcuts that goes beyond solely relying on noisy operations. Through empirical observation of low-rank matrix decomposition in LoRA (Hu et al., 2022) (or the encoder-decoder structure in Auto-Encoder (He et al., 2022)), we identified two inherent properties particularly suitable for addressing the identity mapping issue.

**Property 1** stems from the constrained low-rank latent space, which effectively circumvents identity shortcuts. In deep learning, nonlinear activations are generally used to improve performance, so we

need to prove Property 1 under non-linear transformations, primarily using the Jacobian matrix (Goodfellow et al., 2016) for this proof.

To achieve identity mapping of $x \in \mathbb{R}^d$, we require that $g \circ f(x) = x$ holds for all $x \in \mathbb{R}^n$. Differentiating both sides with respect to $x$ yields the Jacobian equation:

$$J_g(f(x)) \cdot J_f(x) = I_{d \times d} \tag{1}$$

where $J_f(x) \in \mathbb{R}^{d \times k}$ and $J_g(f(x)) \in \mathbb{R}^{k \times d}$. Taking the determinant of both sides:

$$r((J_g(f(x)) \cdot J_f(x))) = r(I_{d \times d}) = d. \tag{2}$$

Since the product $J_g(f(x)) \cdot J_f(x)$ is an $d \times d$ matrix, applying the rank inequality yields:

$$r\left(J_g(f(x)) \cdot J_f(x)\right) \leq \min\left(r(J_g(f(x))), r(J_f(x))\right) \leq k \tag{3}$$

Here, $r(\cdot)$ indicate the rank of the input. Under the low-rank adaptation module, we have $k < d$, Substituting into Eq. equation 3 implies:

$$r\left(J_g(f(x)) \cdot J_f(x)\right) \leq k < d \tag{4}$$

This contradicts Eqs. equation 2 that $det(I_{d \times d})$ requires rank=d. Consequently, no such functions $f$ and $g$ exist.

**Property 2** This property emerges from training exclusively on normal samples, during which the learnable $g$ and $f$ layers are optimized to specifically enhance the reconstruction of normal patterns. This optimization process minimizes adverse effects on the reconstruction fidelity of normal cases, thereby achieving a more favorable trade-off between the reconstruction of normal and abnormal samples.

Specifically, we adapt the LoRA framework (Hu et al., 2022) by implementing Multi-Layer Perceptrons (MLPs) with nonlinear activations to parameterize functions $g$ and $f$, constructing an auto-encoder-like structure. As illustrated in Fig. 3, the modules $g$ and $f$ exhibit symmetrical architectures, where $g$ comprises $i$ downsampling blocks (DBs) and $f$ contains $i$ upsampling blocks (UBs). Within DBs, the output token count of the $i$-th block is halved relative to the $(i-1)$-th block, while UBs correspondingly double the token count relative to their preceding block. Furthermore, we explore integrating feature-level noisy operations via Dropout (Guo et al., 2025), and introduce additional perturbations for better performance.

### 3.3 GLOBAL PERTURBATION ATTENTION (GPA)

The self-attention mechanism in vanilla Transformer inherently facilitates identity mapping issues (You et al., 2022), as it permits unrestricted interaction between feature tokens and their own representations. Therefore, we propose a Global Perturbation Attention (GPA) in the decoder to prevent shortcut pathways in ViT architectures. As illustrated in Fig. 3, GPA comprises two core components: (1) global redistribution and (2) joint global-self masking.

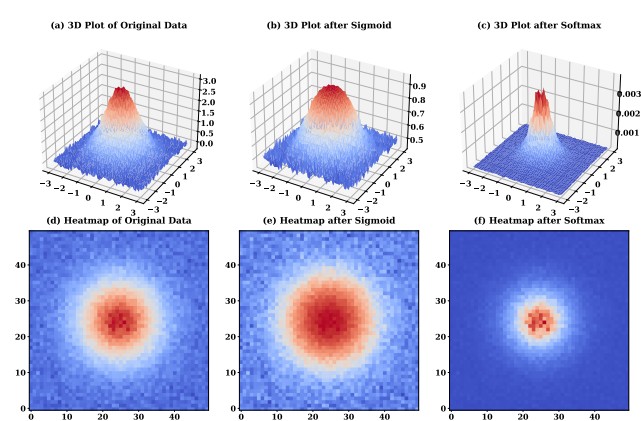

**Global Redistributing (GRD)** In the vanilla self-attention block (Eq. (5)), Softmax function is applied to the query-key attention map to conduct non-linear normalization (Alexey, 2020).

Figure 4: Comparison of sigmoid and softmax Functions on a 2D attention map with Gaussian distribution, (a), (b), (c) are the 3D plot of original data, after softmax and sigmoid respectively, (d), (e), (f) are the corresponding heatmaps.

$$\text{Attention}(Q, K, V) = \text{softmax}\left(\frac{QK^\top}{\sqrt{d_k}}\right) V \tag{5}$$

where $Q \in \mathbb{R}^{N \times d}$, $K \in \mathbb{R}^{N \times d}$ and $V \in \mathbb{R}^{N \times d}$ indicate the query, key and value vectors respectively. $K^T \in \mathbb{R}^{d \times N}$ is the transpose matrix of $K$ and $d_k$ is the scaling factor.

However, as shown in Fig. 4 ((a) vs. (c) and (d) vs. (f)), we observe that the Softmax function tends to overfocus on dominant activation regions during output generation. This behavior is prone to inducing identity mapping under the MUAD setting (Guo et al., 2025). To avoid the overfocusing behavior, the most intuitive approach is to remove Softmax or replace it with an alternative, the Sigmoid (Eq. (6)).

$$\text{Attention}(Q, K, V) = \text{sigmoid}\left(\frac{QK^\top}{\sqrt{d_k}}\right) V \qquad (6)$$

As shown in the 1st and 2nd columns of Fig. 4, when Softmax is removed, the overconcentration issue is alleviated. When Softmax is replaced with the Sigmoid function, the attention spreads more broadly. This indicates that Sigmoid functions tend to leverage longer-range dependencies for feature reconstruction rather than relying on focal attention, thereby reducing the likelihood of propagating unseen information to subsequent layers. Therefore, we apply Sigmoid as a replacement for Softmax in the final setting.

**Global-Self Masking (GSM)** Following global redistribution, GPA applies a global-self-masking operation to the attention map to suppress shortcut learning. The self-masking mechanism effectively prevents tokens from attending to their own positions.

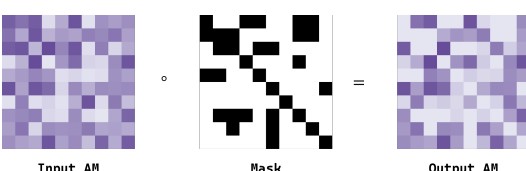

Moreover, considering that feature tokens are captured by the ViT structure, all tokens in the same sequence may share similar information. Therefore, we propose to mask tokens globally and randomly. Specifically, we implement self-masking by masking the elements on the diagonal of the attention map (AM) and conduct random global masking via attention dropout. The flowchart of GSM is shown in Fig. 5; the output AM is obtained by element-wise multiplication of the input AM and our mask.

Figure 5: The flowchart of Global-Self Masking operation. AM: attention map and (·) indicate element-wise multiplication.

## 4 EXPERIMENTS

### 4.1 EXPERIMENTAL SETTING

**Datasets** We evaluated our method on four well-established datasets, including three for industrial scenarios: **MVTec-AD** (Bergmann et al., 2019), **ViSA** (Zou et al., 2022), **Real-IAD** (Wang et al., 2024), and one for medical scenario: **Universal Medical** (He et al., 2024b;a; Zhang et al., 2023; 2024a). **MVTec-AD** comprises 15 object categories, with 3,629 normal images in the training set and 498 normal images alongside 1,982 anomalous samples in the test set. **ViSA** contains 12 categories, providing 8,659 normal training images and a test set with 962 normal images and 1,200 anomalous cases. **Real-IAD**, the largest industrial benchmark, includes 30 diverse objects, utilizing 36,645 normal images for training and 63,256 normal images combined with 51,329 anomalous instances for testing. **Universal Medical** consists of 13,339 normal cases for training, with a test set containing 2,514 normal cases and 4,499 abnormal cases. This dataset spans three medical imaging modalities: Brain MRI, Liver CT, and Retinal CT scans. All datasets include both image-level and pixel-level labels.

**Metrics** For evaluation metrics, we followed protocols from (Zhang et al., 2023; He et al., 2024a; Guo et al., 2025), adopting four metrics: Area Under the Receiver Operating Curve (AUC), F1-max score, average precision (AP) for both image-level detection and pixel-level localization tasks, and Area Under the Per-Region-Overlap (AUPRO) to further evaluate the localization task.

**Implementation details** In our experiments, the encoder is initialized with DINO pre-trained weights (Darcet et al., 2023) and kept frozen during training. Images are resized to 512×512 and then center-cropped to 448×448. We use a stable variant of the AdamW optimizer (Wortsman et al.,

Table 1: Comparison between our method and currently state-of-the-art methods on four datasets, MVTec-AD (Bergmann et al., 2019), ViSA (Zou et al., 2022), Universal Medical (He et al., 2024b) and Real-IAD (Wang et al., 2024). Bold values indicate the best, and underlined values indicates the second best. (%)

| Method | MVTec-AD | | | | | | | ViSA | | | | | | |
| --- | --- | --- | --- | --- | --- | --- | --- | --- | --- | --- | --- | --- | --- | --- |
| | Image-Level | | | Pixel-Level | | | | Image-Level | | | Pixel-Level | | | |
| | AUC | AP | F1 | AUC | AP | F1 | AUPRO | AUC | AP | F1 | AUC | AP | F1 | AUPRO |
| UniAD | 96.5 | 98.8 | 96.2 | 96.8 | 43.4 | 49.5 | 90.7 | 88.8 | 90.8 | 85.8 | 98.3 | 33.7 | 39.0 | 85.5 |
| Recontrast | 98.3 | 99.4 | 97.6 | 97.1 | 60.2 | 61.5 | 93.2 | 95.5 | 96.4 | 92.0 | 98.5 | 47.9 | 50.6 | 91.9 |
| DiAD | 97.2 | 99.0 | 96.5 | 96.8 | 52.6 | 55.5 | 90.7 | 86.8 | 88.3 | 85.1 | 96.0 | 26.1 | 33.0 | 75.2 |
| ViTAD | 98.3 | 99.4 | 97.3 | 97.7 | 55.3 | 58.7 | 91.4 | 90.5 | 91.7 | 86.3 | 98.2 | 36.6 | 41.1 | 85.1 |
| InvAD | 98.9 | 99.6 | 97.2 | 98.1 | 57.2 | 59.5 | 94.3 | 95.5 | 95.8 | 92.1 | 98.9 | 43.1 | 47.0 | 92.5 |
| MambaAD | 98.6 | 99.6 | 97.8 | 97.7 | 56.3 | 59.2 | 93.1 | 94.3 | 94.5 | 89.4 | 98.5 | 39.4 | 44.0 | 91.0 |
| Dinomaly | 99.6 | 99.8 | 99.0 | 98.4 | 69.3 | **69.2** | 94.8 | 98.7 | 98.9 | **96.2** | 98.7 | **53.2** | 55.7 | **94.5** |
| **Ours** | **99.8** | **99.9** | **99.5** | 98.4 | **69.4** | 68.9 | **95.2** | **98.9** | **99.1** | **96.2** | **99.0** | 53.1 | **56.2** | **94.5** |

| Method | Universal Medical | | | | | | | Real-IAD | | | | | | |
| --- | --- | --- | --- | --- | --- | --- | --- | --- | --- | --- | --- | --- | --- | --- |
| | Image-Level | | | Pixel-Level | | | | Image-Level | | | Pixel-Leve | | | |
| | AUC | AP | F1 | AUC | AP | F1 | AUPRO | AUC | AP | F1 | AUC | AP | F1 | AUPRO |
| UniAD | 78.5 | 75.2 | 76.6 | 96.4 | 37.6 | 40.2 | 85.0 | 83.0 | 80.9 | 74.3 | 97.3 | 21.1 | 29.2 | 86.7 |
| Recontrast | 80.1 | 79.7 | 80.9 | 96.3 | 42.3 | 43.8 | 85.2 | 86.4 | 84.2 | 77.4 | 97.8 | 31.6 | 38.2 | 91.8 |
| DiAD | 85.1 | 84.5 | 81.2 | 95.9 | 38.0 | 35.6 | 85.4 | 75.6 | 66.4 | 69.9 | 88.0 | 2.9 | 7.1 | 58.1 |
| ViTAD | 82.2 | 81.0 | 80.1 | 97.1 | 49.9 | 49.6 | 86.1 | 82.3 | 79.4 | 73.4 | 96.9 | 26.7 | 34.9 | 84.9 |
| InvAD | 82.2 | 79.6 | 80.6 | **97.3** | 47.5 | 47.1 | **89.6** | 89.0 | 86.4 | 79.6 | 98.4 | 30.7 | 37.6 | 91.9 |
| MambaAD | 83.7 | 80.1 | 82.0 | 96.9 | 45.4 | 47.3 | 87.5 | 86.3 | 84.6 | 77.0 | 98.5 | 33.0 | 38.7 | 90.5 |
| Dinomaly | 84.9 | 84.1 | 81.0 | 96.8 | 51.7 | 52.1 | 85.5 | 89.3 | 86.8 | 80.2 | 98.8 | 42.8 | 47.1 | 93.9 |
| **Ours** | **87.8** | **87.8** | **82.5** | 97.1 | **54.8** | **54.0** | 87.1 | **90.6** | **87.9** | **81.3** | **99.1** | **44.7** | **49.1** | **95.6** |

2023) incorporating the AMSGrad algorithm (Reddi et al., 2019), with a batch size of 32. Training iterations are set to 20,000 for MVTec-AD and ViSA, 25,000 for Universal Medical, and 50,000 for the largest dataset, Real-IAD. The learning rate is initialized to 2e-3 and gradually reduced to 2e-4 via a Cosine Annealing schedule with a warm-start scheme (Loshchilov & Hutter, 2016). The model is optimized using the global hard-mining loss (Guo et al., 2025). In LRNB, we set the noise rate to 0.1, and set the number of LRNB layers to 2 for MVTec-AD and ViSA, and 3 for Universal Medical and Real-IAD.

## 4.2 EVALUATIONS

**Comparison with SOTA MUAD methods** We evaluate our method against seven state-of-the-art (SOTA) MUAD methods—UniAD (You et al., 2022), DiAD (He et al., 2024b), ViTAD (Zhang et al., 2023), InvAD (Zhang et al., 2024b), Recontrast (Guo et al., 2024), and Dinomaly (Guo et al., 2025)—across four benchmark datasets: MVTec-AD (Bergmann et al., 2019), ViSA (Zou et al., 2022), Universal Medical (He et al., 2024b), and Real-IAD (Wang et al., 2024). Performance is quantified using both image-level metrics (I-AUC, I-AP, I-F1) and pixel-level metrics (P-AUC, P-AP, P-F1, P-AUPRO), where higher values indicate better detection capability. Experimental results are presented in Table 1, where our method outperforms comparative methods across all datasets in most metrics (The qualitative results obtained by our method can be seen in Appendix 3). On the widely adopted MVTec-AD, our method achieves SOTA overall performance, with the highest image-level metrics of **99.8/99.9/99.5**, as well as three top-ranked and one second-ranked pixel-level metrics of **98.4/69.4**/68.9/**95.2**. On ViSA, our method consistently achieves the best image-level performance of **98.9/99.1/96.2** and competitive pixel-level performance of **99.0**/53.1/**56.2/94.5**. These results demonstrate that image-level performance on these two datasets is nearly 100On Universal Medical, our method attains the highest image-level metrics of **87.8/87.8/82.5**, surpassing prior SOTAs by a large margin of 2.6/3.4/0.5, and achieves best or third-best pixel-level metrics of 97.1/**54.8/54.0**/87.1. On Real-IAD, our method produces a new SOTA result, with image-level and pixel-level performance of **90.6/87.9/81.4** and **99.1/44.7/49.1/95.6**, outperforming previous SO-

Table 2: Ablations studies of component contributions on Universal Medical dataset, including LRNB: low-rank noisy bottleneck, GRD: global redistribution operation and GSM: global-self masking mechanism. GPA: global perturbation attention, which is the combination of GRD and GSM. (%)

| LRNB | GPA | | Image-level | | | Pixel-level | | | |
|---|---|---|---|---|---|---|---|---|---|
| | GRD | GSM | AUC | AP | F1 | AUC | AP | F1 | AUPRO |
| | | | 79.28 | 78.67 | 80.41 | 95.49 | 39.30 | 41.11 | 81.35 |
| ✓ | | | 86.38 | 85.72 | 81.67 | 96.83 | 51.83 | 52.29 | 86.18 |
| | ✓ | | 79.89 | 78.60 | 80.26 | 95.70 | 41.03 | 42.11 | 81.77 |
| | | ✓ | 79.48 | 78.77 | 80.50 | 95.53 | 39.54 | 41.40 | 81.78 |
| ✓ | ✓ | | 87.53 | 86.96 | 82.42 | 97.05 | 54.74 | 53.89 | 87.11 |
| ✓ | | ✓ | 86.70 | 87.12 | 81.53 | 96.98 | 53.43 | 52.88 | 86.09 |
| ✓ | ✓ | ✓ | **87.81** | **87.75** | **82.45** | **97.06** | **54.81** | **54.03** | 87.06 |

TAs by 1.3/1.1/1.1 and 0.3/1.9/2.0/1.7. These results demonstrate strong generalization to complex medical and diverse real-world industrial scenarios.

## 4.3 Ablation Studies

**Overall Ablation** To explore the contributions of each component in our method, including the low-rank noisy bottleneck (LRNB), global redistribution (GRD), and global-self masking (GSM) operations, we conduct ablation experiments as shown in Table 2. The baseline is constructed following Dinomaly (Guo et al., 2025) and ViTAD (Zhang et al., 2023), which construct a baseline model with a DINO-pretrained ViT encoder and a learnable softmax attention-based ViT decoder. The effectiveness of the baseline has been proven in industrial scenarios. The results of the ablation experiments demonstrate that the proposed LRNB, GRD, and GSM modules all contribute to performance improvement, with their combinations further enhancing performance. Compared to the baseline model, using any single module alone achieves better performance in terms of the key indicators I-AUC and P-AUC. Among them, GRD and GSM bring moderate improvements, while LRNB shows the most significant enhancement—it notably boosts image-level and pixel-level performances with respective large margins of 7.10/7.05/1.26 and 1.44/12.53/11.18/4.83. These results demonstrate that the LRNB module serves as a core foundation in our method. Moreover, module combinations yield even better results than single modules. When LRNB is combined with either GRD or GSM, performance exceeds that of LRNB used alone. The combination of LRNB and GRD stands out particularly in optimizing I-AUC and P-AUC. The integration of all three modules (LRNB + GRD + GSM) delivers the best overall performance, achieving the highest image-level metrics **87.81/87.80/82.45** and top/top-2 ranked pixel-level performance **97.06/54.81/54.03**/87.06—confirming that their collaborative effect effectively enhances the model's overall performance.

| BN | Image-level | | | Pixel level | | | |
|---|---|---|---|---|---|---|---|
| | AUC | AP | F1 | AUC | AP | F1 | AUPRO |
| None | 78.6 | 77.7 | 80.8 | 95.3 | 38.6 | 40.1 | 81.2 |
| FJ | 77.7 | 74.0 | 80.2 | 95.6 | 39.5 | 42.3 | 81.1 |
| NDB | 82.8 | 81.8 | 80.4 | 96.3 | 44.6 | 46.0 | 83.9 |
| LRNB | **87.8** | **87.8** | **82.5** | **97.1** | **54.8** | **54.0** | **87.1** |

Table 3: Performance comparison of different bottlenecks (BNs) on Universal Medical. (%)

| Modules | Image-level | | | Pixel level | | | |
|---|---|---|---|---|---|---|---|
| | AUC | AP | F1 | AUC | AP | F1 | AUPRO |
| ViT | 86.4 | 85.7 | 81.7 | 96.8 | 51.8 | 52.3 | 86.2 |
| CNN | 84.6 | 84.9 | 81.3 | 96.4 | 48.4 | 48.9 | 85.1 |
| CNN+ViT | 83.9 | 83.8 | 81.2 | 96.4 | 47.8 | 48.4 | 84.1 |
| NMA | 87.4 | 86.6 | 82.0 | 97.0 | 53.2 | 53.6 | 86.4 |
| GPA | **87.8** | **87.8** | **82.5** | **97.1** | **54.8** | **54.0** | **87.1** |

Table 4: Performance comparison of different modules in decoder on Universal Medical. (%)

**Comparison between LRNB and previous noisy bottlenecks** Previous methods also proposed some noisy operations in the bottleneck to avoid identity shortcuts, such as feature jittering (FJ) in UniAD (You et al., 2022) and dropout noisy bottleneck (DNB) in Di-

nomaly (Guo et al., 2025). To further validate the advantage of the proposed LRNB, we conduct a comparison in Table 3. The results in this table demonstrate that FJ experiences a performance drop and DNB yields slight improvements compared to models without such noise operations, while our LRNB achieves a significant enhancement.

To explore the capacity of these operations in addressing the "identity mapping" issue, we further examine the corresponding training loss and average anomaly score. As shown in Fig. 6, models without a bottleneck achieve near-zero training loss and obtain extremely close anomaly scores between normal and abnormal cases (0.0195 vs. 0.0223). These results indicate that these models suffers from identity mapping behavior where inputs are directly replicated in outputs. Introducing FJ or DNB noise operations partially alleviates this issue, which we attribute to en-

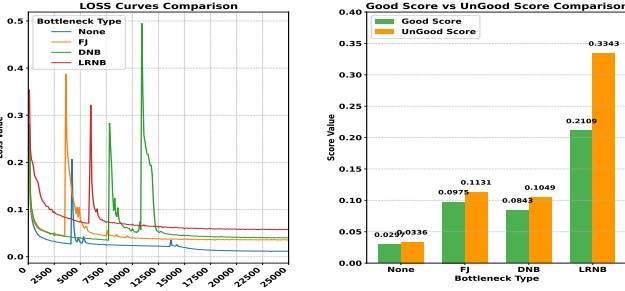

Figure 6: The plots of loss (left part) and the averaged anomaly scores (right part) of different bottlenecks on Universal Medical.

hanced noise robustness from exposure to diverse medical patterns. Our LRNB framework effectively resolves identity mapping: while moderately increasing the normal score, it substantially enhances the abnormal score. This strategic trade-off widens the discrimination margin between normal and abnormal samples, ultimately improving anomaly detection performance.

**Comparison between GPA and previous modules in decoder** Prior works proposed decoder modules to address the identity mapping issue. (Lu et al., 2024) attributes it to encoder-decoder homogeneity and advocates heterogeneous decoders for reconstructing encoder outputs (e.g., using ViT as encoder and CNN as decoder blocks). (You et al., 2022) introduced neighbor-masking attention (NMA) to prevent information leakage from CNN-extracted tokens. We compared ViT (baseline), CNN, CNN+ViT, NMA, and our GPA in Table 4. The table shows that replacing ViT with CNN or CNN+ViT yields poorer performance, as CNNs are more prone to identity mapping (You et al., 2022). Fig. 7 in Appendix 2.1 supports this: training losses of CNN and CNN+ViT drop sharply—even lower than ViT's—narrowing the score distance between normal and abnormal cases. In contrast, NMA and our GPA outperform ViT, where the corresponding increased training losses and reconstruction errors for both cases confirm they alleviate shortcut learning. Finally, our GPA outperforms NMA in most metrics, which we attribute to its adaptability to ViT-extracted tokens.

## 5 CONCLUSION

In this paper, we propose ShortcutBreaker, a novel feature-reconstruction framework designed to mitigate the identity shortcut issue in the MUAD setting. It incorporates two core innovations: a low-rank noisy bottleneck and a global perturbation attention mechanism, which significantly enhance performance by preventing shortcut learning in the bottleneck and decoder components. Extensive experiments confirm the efficacy of these components. Furthermore, the state-of-the-art performance achieved on four MUAD benchmarks (MVTec-AD, ViSA, Real-IAD, and Universal Medical) demonstrates consistent superiority over prior methods, particularly in complex and diverse scenarios.

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
