# SHORTCUTBREAKER: LOW-RANK NOISY BOTTLE-NECK WITH GLOBAL PERTURBATION ATTENTION FOR MULTI-CLASS UNSUPERVISED ANOMALY DETECTION

## ABSTRACT

Multi-class unsupervised anomaly detection (MUAD) has garnered growing research interest, as it seeks to develop a unified model for anomaly detection across multiple classes—eliminating the need to train separate models for distinct objects and thereby saving substantial computational resources. Under the MUAD setting, while advanced Transformer-based architectures have brought significant performance improvements, identity shortcuts persist: they directly copy inputs to outputs, narrowing the gap in reconstruction errors between normal and abnormal cases, and thereby making the two harder to distinguish. Therefore, we propose ShortcutBreaker, a novel unified feature-reconstruction framework for MUAD tasks, featuring two key innovations to address the issue of shortcuts. First, drawing on matrix rank inequality, we design a low-rank noisy bottleneck (LRNB) to project high-dimensional features into a low-rank latent space, and theoretically demonstrate its capacity to prevent trivial identity reproduction. Second, leveraging ViT's global modeling capability instead of merely focusing on local features, we incorporate a global perturbation attention to prevent information shortcuts in the decoders. Extensive experiments are performed on four widely used anomaly detection benchmarks, including three industrial datasets (MVTec-AD, ViSA, and Real-IAD) and one medical dataset (Universal Medical). The proposed method achieves a remarkable image-level AUROC of *99.8%, 98.9%, 90.6%*, and *87.8%* on these four datasets, respectively, consistently outperforming previous MUAD methods across different scenarios. Our code will be released.

## 1 INTRODUCTION

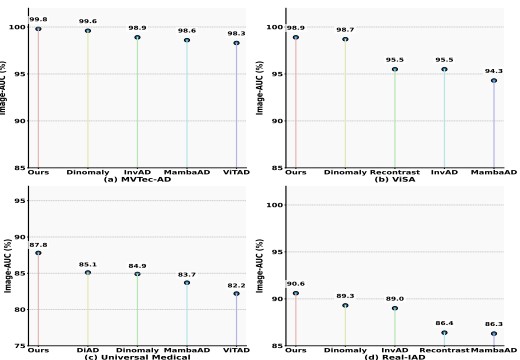

Figure 1: Performance comparison in terms of image-level AUC on MVTec, ViSA , Universal Medical and Real-IAD .

Anomaly detection in industrial and medical imaging aims to identify abnormal patterns among normal cases, saving labor and time in collecting and labeling anomalies. Given the abundance of normal cases and the scarcity of anomalies, this task is typically tackled via an unsupervised paradigm using only normal training samples. Before deep learning, anomaly detection relied on traditional techniques: density-based methods (Breunig et al., 2000; Guan et al., 2015), distance-based methods (Knorr et al., 2000; Angiulli et al., 2005), and statistics-based methods (Hido et al., 2011; Rousseeuw & Hubert, 2011). Current state-of-the-art methods employ deep learning networks pre-trained on ImageNet (Deng et al., 2009) to capture discriminative features. Feature reconstruction methods (Guo et al., 2023; 2024; Deng & Li, 2022) reconstruct encoder-extracted features, assuming accurate reconstruction of normal regions but failure on unseen anomalies. Memory matching methods (Yi & Yoon, 2020; Defard

et al., 2021; Roth et al., 2022) memorize training-set normal features for inference matching, with those in (Defard et al., 2021; Roth et al., 2022) using pre-trained encoders for discriminative features. Pseudo-anomaly methods (Liu et al., 2023; Li et al., 2021; Zavrtanik et al., 2021) convert UAD to a supervised task by generating pseudo anomalies via noise addition to normal images/features. Hybrid methods (Tien et al., 2023; Zhao et al., 2023) integrate normalizing flows (Zhao et al., 2023) or pseudo noise (Tien et al., 2023) into feature reconstruction for UAD.

Despite their success, these methods are limited to a one-model-one-class setup, requiring substantial storage for per-class models—especially with many disease types (You et al., 2022). To address this, UniAD and follow-ups propose unified models for multi-class unsupervised anomaly detection (MUAD). However, identity mapping often emerges here, where the model returns input copies regardless of normality, enabling effective reconstruction of even anomalous samples and hindering detection (You et al., 2022).

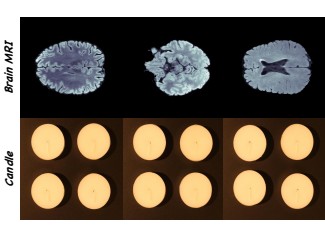 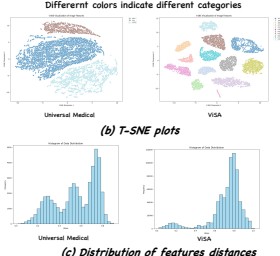

(a) Visualization of medical and industrial images

(b) T-SNE plots

(c) Distribution of features distances

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

# A APPENDIX

## A.1 PROOFS

In this section, we will provide the proof of the first property of low-rank matrix decomposition under non-linear transformation, the analysis of rank constraints of the proposition 1 in Cai et al. (2024). In our proofs, we assume the input feature matrix $x \in \mathbb{R}^d$ maintains full column rank. This assumption aligns with our empirical observations from practical computations using Singular Value Decomposition (SVD).

**Analysis of rank constraints of the proposition 1 in Cai et al. (2024)** Given the condition of linear transformation, Proposition 1 in Cai et al. (2024) establishes that for $k \geq \frac{d}{2}$, the $W_1 W_2 = I_{d \times d}$ with $W_1 \in \mathbb{R}^{d \times k}$ and $W_2 \in \mathbb{R}^{k \times d}$ admits at least one solution. Consequently, in turns, when no solutions exist for this equation, $k < \frac{d}{2}$, thereby inherently suppressing identity mapping. However, a critical paradox emerges in the transitional regime $d > k \geq \frac{d}{2}$: Proposition 1 guarantees solution existence, but the rank inequality $r(W_1 W_2) \leq min(r(W_1), r(W_2)) \leq k < d$ directly conflicts with the full-rank requirement $I_{d \times d} = d$. Therefore, this reasoning process is flawed.

## A.2 ADDITIONAL ABLATION STUDIES

**More results between GPA and previous modules in decoder** Fig. 7 displays the training losses and averaged anomaly scores of different modules in decoder. This figure further supports this finding: the training losses of CNN and CNN+ViT decrease sharply—even lower than those of ViT—thereby narrowing the score distance between normal and abnormal cases. In contrast, NMA and our GPA outperform ViT. Their increased training losses and reconstruction errors for both case types demonstrate that these two modules effectively alleviate shortcut learning. Finally, our GPA achieves higher values in most metrics than NMA; we attribute this to the adaptability of our proposed GPA to tokens extracted by ViT.

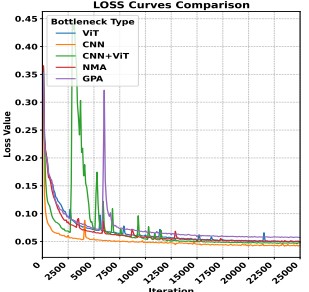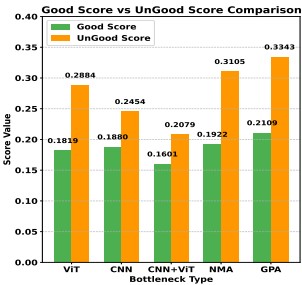

Figure 7: The plots of loss (left part) and the averaged anomaly scores (right part) of different modules in decoder on Universal Medical. CNN: convolution neural network, ViT: vision transformer, NMA: neighbor-masking attention and GPA: global perturbation attention.

**Further analysis in LRNB** As shown in Fig. 8, the results indicate that when the number of layers exceeds three, the performance stabilizes. Although a larger number of layers can further increase the anomaly scores for both types of samples, once the number exceeds three, the gap between normal and defective cases does not widen, leading to stable performance. Additionally, Table 5 presents the effects of different dropout rates in the LRNB model. As shown in this table, introducing noise operations improves overall performance within the range of 0.1 to 0.4; specifically, when the dropout rate is set to 0.1, the model achieves the highest values in most metrics.

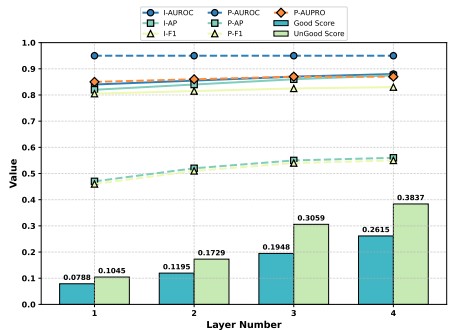

Figure 8: The plots of metrics (left part) and the averaged anomaly scores (right part) of different layer number in LRNB. Influences of the number of layer in LRNB on Universal Medical.

Table 5: Effect of noise rate of LRNB on Universal Medical. (%)

| Noise rate | Image-level | | | Pixel level | | | |
|---|---|---|---|---|---|---|---|
| | AUC | AP | F1 | AUC | AP | F1 | AUPRO |
| 0.0 | 86.2 | 86.2 | 81.8 | 97.0 | 52.1 | 52.1 | 86.5 |
| 0.1 | **87.8** | **87.8** | **82.5** | **97.1** | **54.8** | **54.0** | 87.1 |
| 0.2 | 87.6 | 87.5 | 82.3 | 97.0 | 53.4 | 53.1 | 87.1 |
| 0.3 | 87.2 | 86.9 | 82.4 | 96.9 | 54.1 | 53.4 | **87.4** |
| 0.4 | 87.7 | 87.3 | 82.6 | 97.0 | 54.3 | 53.6 | 87.3 |

Table 6: Effect of attention dropout rate in GSM in Universal Medical, Attndrop: Attention Dropout. (%)

| Attndrop rate | Image-level | | | Pixel level | | | |
|---|---|---|---|---|---|---|---|
| | AUC | AP | F1 | AUC | AP | F1 | AUPRO |
| 0.0 | 87.6 | 87.5 | 82.2 | 97.0 | 54.1 | 53.4 | 87.0 |
| 0.1 | 87.3 | 87.4 | 82.5 | 97.0 | 54.8 | 54.0 | 87.0 |
| 0.2 | 87.6 | 87.9 | 82.5 | 97.0 | 53.9 | 53.2 | 86.9 |
| 0.3 | **87.8** | **87.8** | **82.5** | **97.1** | 54.8 | 54.0 | 87.1 |
| 0.4 | 87.4 | 87.1 | 82.4 | **97.1** | **55.2** | **54.3** | **87.3** |

**Effect of different attention dropout rate in GSM** The results in Table 6 illustrate the effects of different attention dropout rates in the GSM model. As shown in this table, when the rate is set to 0.3, the model achieves optimal overall performance at the image-pixel level. Accordingly, based on the empirical settings of Dinomaly Guo et al. (2025), we set the attention dropout rate to 0.3 for Universal Medical and Real-IAD, and to 0.2 for the relatively simpler MVTec-AD and ViSA datasets.

### A.3 QUALITATIVE RESULTS

In this section, we further visualize the anomaly maps of our ShortcutBreaker, on four datasets. Fig. 9-12 presents the visualization results of MvTec-AD Bergmann et al. (2019), ViSA Zou et al. (2022), Universal Medical He et al. (2024b) and Real-IAD Wang et al. (2024) datasets, respectively.

## B REPRODUCIBILITY STATEMENT

We have already elaborated on all the models or algorithms proposed, experimental configurations, and benchmarks used in the experiments in the main body or appendix of this paper. Furthermore, we declare that the entire code used in this work will be released after acceptance.

## C THE USE OF LARGE LANGUAGE MODELS

We use large language models solely for polishing our writing, and we have conducted a careful check, taking full responsibility for all content in this work.

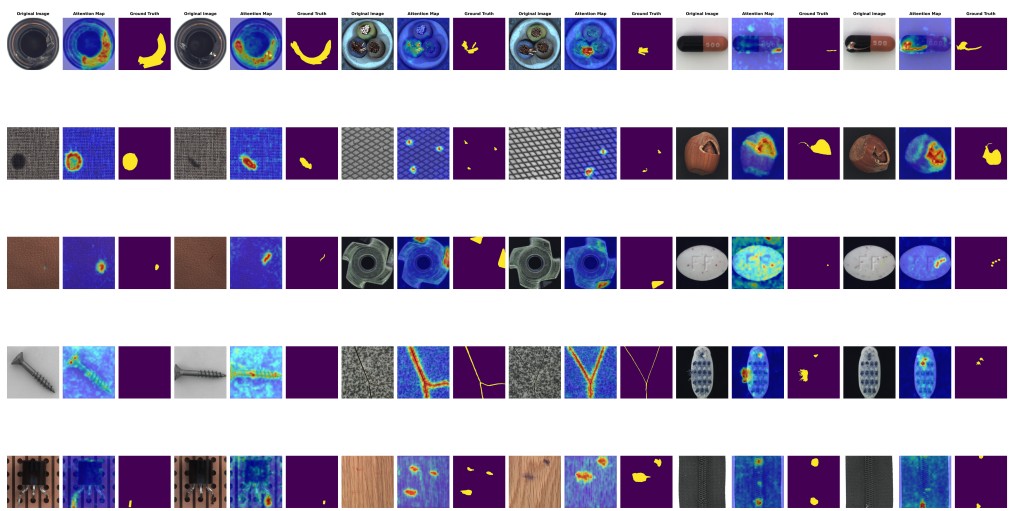

Figure 9: Qualitative results of our method on MVTec-AD Bergmann et al. (2019).

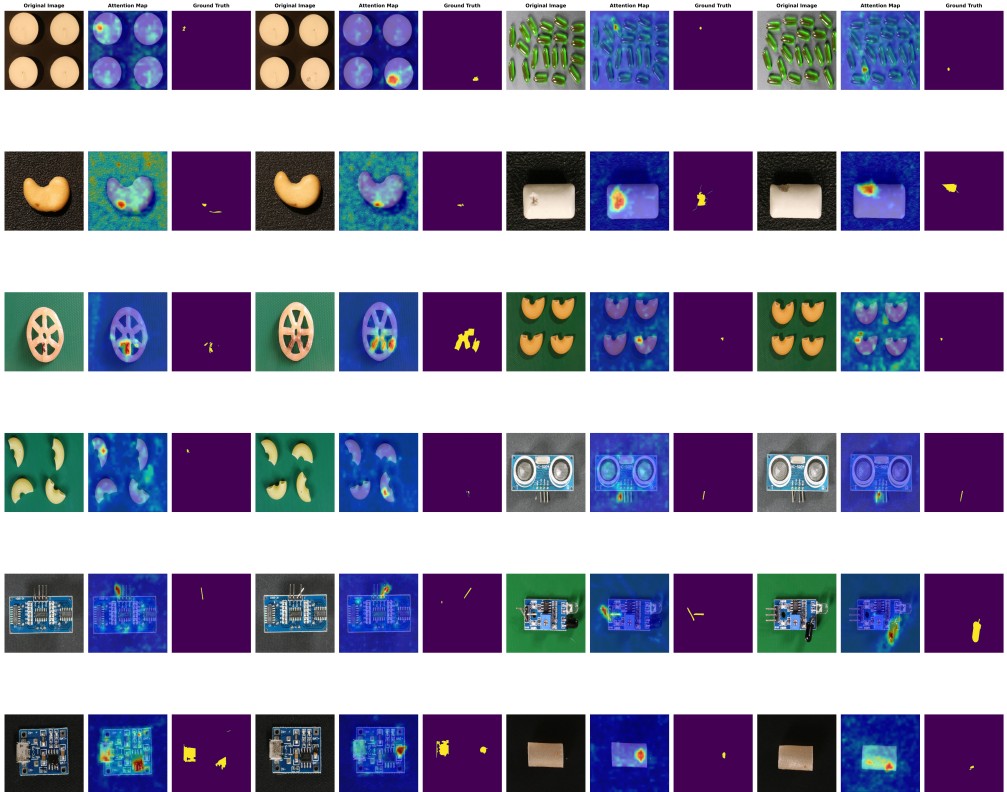

Figure 10: Qualitative results of our method on ViSA Zou et al. (2022).

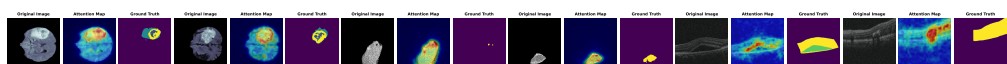

Figure 11: Qualitative results of our method on Universal Medical He et al. (2024b).

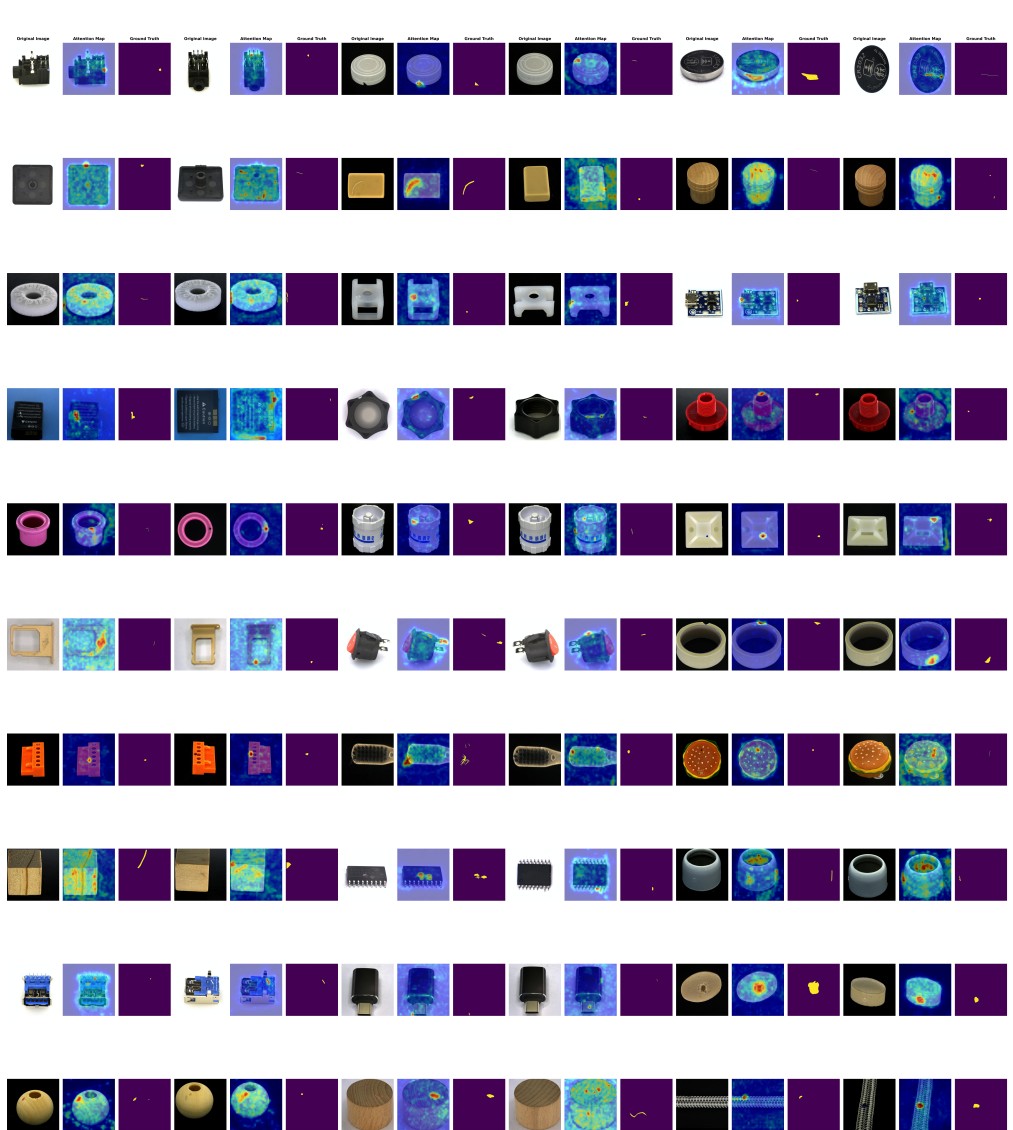

Figure 12: Qualitative results of our method on Real-IAD Wang et al. (2024).