# OpenReview forum: "ShortcutBreaker: Low-Rank Noisy Bottleneck with Global Perturbation Attention for Multi-Class Unsupervised Anomaly Detection"
_ICLR.cc/2026/Conference — ICLR 2026 Conference Withdrawn Submission_

### Official Review · Reviewer_tqJw · 2025-10-29

**Soundness:** 3
**Presentation:** 2
**Contribution:** 3
**Rating:** 4
**Confidence:** 4

**Summary:**

This paper proposes a novel Multi-Class Unsupervised Anomaly Detection (MUAD) method called ShortcutBreaker, which addresses the widespread identity shortcut problem in multi-class reconstruction models—where the model tends to directly copy input features, making it difficult to distinguish between normal and anomalous samples. The key innovations include the Low-Rank Noisy Bottleneck (LRNB) and the Global Perturbation Attention (GPA). The authors conduct systematic experiments on four public datasets: MVTec-AD, ViSA, Real-IAD, and Universal Medical. Results show that the proposed method outperforms existing approaches on all datasets, with particularly significant improvements on the complex Real-IAD and medical datasets. Ablation studies further validate the effectiveness and synergistic benefits of each module.

**Strengths:**

1.The use of matrix rank inequalities demonstrates that low-rank constraints can effectively prevent identity mapping, enhancing the theoretical reliability of the method.
2.The combination of LRNB and GPA jointly restricts shortcut pathways from both the encoder and decoder sides, forming a compact and complementary structure.
3.It demonstrates consistent superiority across four representative datasets, maintaining strong performance especially on complex multi-class data.
4.The mitigation of shortcuts is illustrated through loss curve analysis and feature distribution visualization.

**Weaknesses:**

In terms of writing:
The writing details need improvement. For instance, "MatMul" appears twice in Figure 3, and the textual expression is suboptimal, such as in Section 4.2.
In terms of innovation:
1.Although the motivation for replacing Softmax with Sigmoid in GPA is supported by some experiments, there is a lack of theoretical proof of its superiority, and no explanation is provided as to why Sigmoid can alleviate identity mapping.
2.The LRNB assumes that normal samples can be represented by a low-rank structure. However, normal samples in medical images are highly heterogeneous (e.g., differences in organ size and morphology), and the low-rank assumption may not hold, potentially even leading to information loss. The analysis of LRNB's low-rank property is only at the level of Jacobian matrix rank inequalities, lacking a deeper discussion on the stability of the low-rank structure under nonlinear transformations.
3.Although two modules, LRNB and GPA, are proposed, the method essentially still belongs to the framework of "feature reconstruction + attention mechanism", and has not departed from the mainstream paradigm of existing MUAD methods.

**Questions:**

1.How can it be proven that the low-rank assumption holds for all normal samples, especially given the highly heterogeneous anatomical structures present in medical images?
2.Is there a theoretical derivation for the identity-mapping prevention effect of replacing Softmax with Sigmoid, rather than relying solely on empirical observation?
3.Some metrics in Table 1 did not achieve the best performance—could you explain the possible reasons for this?

---

### Official Review · Reviewer_uA31 · 2025-11-01

**Soundness:** 2
**Presentation:** 3
**Contribution:** 1
**Rating:** 2
**Confidence:** 5

**Summary:**

This paper presents ShortcutBreaker to address identity shortcuts in multi-class unsupervised anomaly detection via LRNB and GPA. While it achieves marginal performance gains on benchmarks, critical flaws in design logic, innovation, and theoretical rigor significantly limit its contribution.

**Strengths:**

The method shows modest performance improvements compared to existing baselines across four datasets, demonstrating some practical effectiveness.

**Weaknesses:**

1. Softmax overconcentration has no direct correlation with identity shortcuts. GRD lacks normalization, making the model sensitive to input scales—weights may cluster near 1 (averaging values) or 0 (output collapse), leading to unstable gradients and unfocused attention.
2. The work lacks innovation: LRNB, GRD, and GSM are all derivative of existing structures from prior literature.
3. The understanding of related work is insufficient—NMA is not exclusively designed for CNN features, and noise operations’ effectiveness is unrelated to dataset complexity.
4. The theoretical proof relies on a false assumption that input features have rank d, invalidating its core theoretical argument.

**Questions:**

What is the relationship between GPA Decoder and the LoRA framework? Do you use a pretrained model here?

---

### Official Review · Reviewer_uK2E · 2025-11-02

**Soundness:** 3
**Presentation:** 4
**Contribution:** 3
**Rating:** 6
**Confidence:** 2

**Summary:**

This study proposes a feature reconstruction framework called ShortcutBreaker for Multi-Class Unsupervised Anomaly Detection (MUAD), with the motivation to address the identity shortcut problem in Transformer architectures under the MUAD setting and improve anomaly detection performance. The framework includes two core innovations.  First, the design of a Low-Rank Noise Bottleneck (LRNB), which projects high-dimensional features output by the encoder into a low-rank latent space and introduces noise through dropout to prevent identity reproduction. Second, the integration of Global Perturbation Attention (GPA) by leveraging the global modeling capability of Vision Transformer (ViT), which blocks the information shortcut of the decoder through Global Redistributing (GRD) and Global-Self Masking (GSM) operations.  The method outperforms existing MUAD approaches on four datasets, namely MVTec-AD, ViSA, Real-IAD, and Universal Medical.

**Strengths:**

1), The proposed method effectively blocks the identity shortcut problem in the MUAD reconstruction framework.

2), It outperforms the State-of-the-Art on four benchmark datasets, which cover industrial and medical scenarios, demonstrating strong generalization ability.

**Weaknesses:**

1), There is insufficient qualitative visualization, and the paper lacks parameter sensitivity analysis (e.g., the rank and noise rate in LRNB).

**Questions:**

1), How sensitive is the model’s performance to different noise rates in the LRNB module?

2), What is the computational overhead of the proposed method?

3), Is the random global masking ratio consistent across all layers and datasets?

---

### Official Review · Reviewer_muHh · 2025-11-08

**Soundness:** 3
**Presentation:** 2
**Contribution:** 2
**Rating:** 6
**Confidence:** 4

**Summary:**

This paper focuses on the identity shortcut issue in multi-class unsupervised anomaly detection and proposes a feature-reconstruction method using two components: a low-rank noisy bottleneck (LRNB) and global perturbation attention (GPA). The method achieves strong results on four datasets.

**Strengths:**

* This paper presents a clear motivation for addressing the identity shortcut issue. Two components are closely related to this motivation. The authors also provide a clear illustration of their specific method.
* The promising results demonstrate the effectiveness of the proposed method. Additionally, the authors conducted a comprehensive ablation study to evaluate the contribution of each component and parameter.

**Weaknesses:**

* Property 1 of LRNB is very similar to the idea of VAE. It only rules out global identity functions. It is not clear if there are local approximate identity mappings
* It is not clear why "noisy operations are often limited in effectively preventing shortcut learning in complex scenarios (lines 207-210)." Perhaps the authors should consider comparing their approach with previous techniques for resolving identity shortcut issues, rather than merely presenting results or exclusively.
* The reason for introducing noise in the LRNB decoder is not clarified.
* All techniques, including bottleneck and noise in LRNB, sigmoid function and global mask in GAP, will compress/lose detailed information. Could this pose a problem when detecting very small anomalies, such as small defects? The paper does not analyze these side effects. It’s difficult to determine from the current results (both pixel-level and image-level), since all four datasets contain both large and small anomalies.
* The connection between the two components is not discussed. For example, let's say "the global constraint is derived from LRNB, while the local constraint is based on GPA".
* No discussion about computing resources and time.

**Questions:**

And Other Comments
* Will sigmoid attention lead to issues such as gradient saturation and scale drift?
* Could the authors provide visualizations of intermediate reconstruction before/after LRNB and GPA to validate the claimed mechanism?
* Papers on this topic typically include anomaly score maps. Including such visualizations could partially address weakness 4.
* Some figures are too small to read clearly.

Please note I did not consider this section in the rating. I was going to give 5 and was willing to increase to 6 at least if the authors could address some of my concerns. But since only options 4 and 6 are available, I will give 6 in advance.

---

### Note · Authors · 2025-11-13

**Comment:**

Thanks AC Reviewer muHh, uK2E and tqJw,


We will revise our paper based on your useful suggestions.

**Withdrawal Confirmation:**

I have read and agree with the venue's withdrawal policy on behalf of myself and my co-authors.